# Enhanced Organic Pollutant Removal Efficiency of Electrospun NiTiO_3_/TiO_2_-Decorated Carbon Nanofibers

**DOI:** 10.3390/polym15010109

**Published:** 2022-12-27

**Authors:** Ibrahim M. Maafa, Mohammad Ashraf Ali

**Affiliations:** Department of Chemical Engineering, College of Engineering, Jazan University, Jazan 45142, Saudi Arabia

**Keywords:** electrospinning, nickel titanate, nanofibers, photodegradation, mathematical modeling

## Abstract

A nanocomposite comprised of nickel titanate/titania nanoparticles decorated with carbon nanofibers (NiTiO_3_/TiO_2_-decorated CNFs) is successfully synthesized via electrospinning and further utilized for methylene blue (MB) photodegradation. The morphology, phase, structural and chemical composition of the nanocomposite is investigated via scanning electron microscope, X-ray diffraction and transmission electron microscope equipped with energy dispersive X-ray. A mathematical model is developed to predict the photocatalytic activity of the produced nanocomposite by considering parameters such as initial dye concentration, light intensity, reaction temperature, and catalyst dosage. The reaction rate constant K_1_ decreased from 0.0153 to 0.0044 min^−1^ with an increase in the MB concentration from 5 to 15 mg L^−1^, while K_2_, K_3_, and K_4_ were found to increase with the increase in reaction temperature (0.0153 to 0.0222 min^−1^), light intensity (0.0153 to 0.0228 min^−1^) and catalyst dose concentration (0.0153 to 0.0324 min^−1^), respectively. The results obtained are found to be in good agreement with the modeling results and showed effective photodegradation activity. The performance of our catalyst is found to be better compared to other catalysts previously reported in the literature. The recyclability data of the synthesized NiTiO_3_/TiO_2_-decorated CNFs catalyst for four runs show that the catalyst is quite stable and recyclable. This nanocomposite photocatalyst offers a low-cost solution for wastewater pollution problems and opens new avenues to further explore the electrospinning method for the synthesis of nanocomposites.

## 1. Introduction

Organic pollutants produced from industrial water pose a serious threat to our environment, which will eventually affect the health of humans and aquatic life [1,2,3]. Therefore, solving this problem calls for urgent attention to not only get rid of these pollutants but also to overcome the shortage of usable fresh water. According to the UNESCO World Water Assessment Program (2003), about 2 million tons of disposed water from industrial and agricultural sources is released every day [4]. Thus, recycling and reusing disposed water has become a necessity. However, an effective and safe method to obtain usable water requires a high cost. Traditional methods such as chemical coagulation, flocculation, and adsorption have been employed to overcome the aforementioned issue; however, these methods suffer a major drawback in the formation of a secondary pollutant, and thus require additional steps to remove this, incurring an additional cost [5,6,7,8,9]. The photocatalytic process utilizing semiconducting materials is considered as an eco-friendly and inexpensive method for the decomposition and mineralization of untreated pollutants resulting from industrial wastewater employing ultraviolet or visible light irradiation [10,11]. Several photocatalytic semiconductors are being used in the decomposition of organic and inorganic pollutants. Titanium dioxide (TiO_2_) is the most common semiconductor utilized in the photocatalytic processes such as dye-sensitized solar cells (DSSC), photodegradation of untreated pollutants, self-cleaning paint, and water splitting [12,13,14]. TiO_2_ is a low-cost material which is chemically stable, environmentally compatible, and easily available. However, due to wide gap between its conduction and valance band (Eg~3.0 eV for rutile and Eg~3.2 eV for anatase) and its low quantum yield, it absorbs light in the UV region only, which impedes its practical application in photocatalysis [15,16]. Moreover, the quick recombination of light-induced electron (e^−^) and hole (h^+^) pairs leads to a decrease in the formation of radicals. The addition of metallic or non-metallic oxide to TiO_2_ leads to the formation of a narrow band gap and suppresses the recombination of photo-induced e^−^ and h^+^, thereby shifting the absorption towards the visible spectrum [17,18]. During the past years, perovskite-based titanate has been introduced as an efficient photocatalyst in the visible spectrum [19,20,21]. Moreover, there has been a considerable attention focused on the fabrication of heterojunction structures of perovskite-based titanate and TiO_2_ [19,20,21,22]. Among them, nickel titanate (NiTiO_3_, Eg~2.18 eV) has been introduced as a competent photocatalyst in the degradation of untreated pollutants due to its suitable band gap and ability to absorb sunlight radiation [23]. Huang et al. synthesized NiTiO_3_/TiO_2_ nanotubes and employed them for H_2_ production via the water splitting technique [24]. The synthesized material demonstrated good photocatalytic activity, as 45 and 680 µmol g^−1^ h^−1^ for pristine TiO_2_ nanotube and synthesized material, respectively. Integration of a photocatalyst and adsorption could improve the overall wastewater treatment process. A combination of both processes could enhance the degradation performance of organic pollutants as compared to the individual processes [4]. Activated carbon is the most common adsorbent material and has been applied to various composites as support and an absorbent for photocatalyst material including TiO_2_/AC [25], ZnO/AC [26], and CeO_2_/AC [27]. Due to their unique properties, nano-carbon materials such as carbon nanofibers (CNFs), carbon nanotubes (CNT), and graphene have gained extensive interest and are widely used in various applications [28,29]. Moradi et al. incorporated graphene oxide (GO) with FeTiO_3_ to reduce its band gap and showed an enhanced shift towards the visible spectrum [30]. They demonstrated that the visible light offers faster degradation compared to the UV light. The material was synthesized via an ultrasound-assisted method and showed enhanced photocatalytic activity towards phenolic compound degradation under sunlight after 240 min. Their study reported that the rate of phenol removal was increased to 44% with an increase in addition of GO to FeTiO_3_ from 1 to 3%. GO works as a good e^−^ receiver, which inhibits electron and hole recombination. Furthermore, GO improved phenol adsorption in GO/FeTiO_3_, wherein phenol readily reacts with hydroxyl radicals and holes present on the surface. The previous studies also show the enhanced photocatalytic activity of CNFs towards pollutant adsorption from water due to small pores and high surface area formed during the electrospinning process and calcination [31]. Besides the enhancement of adsorption, CNFs have high electrical conductivity, which promotes the capture and transfer of photo-induced charges during the photosynthesis process [32,33]. This present work reports the synthesis of heterojunction NiTiO_3_/TiO_2_-decorated CNFs and their visible-light-driven MB degradation as a model dye. The photocatalyst was synthesized employing an electrospinning technique followed by calcination at 800 °C of the electrospun nanofiber mats composed of titanium (IV) isopropoxide, nickel acetate tetrahydrate and poly(vinylpyrrolidone).

## 2. Experimental

### 2.1. Materials

Titanium isopropoxide (TiIP, 97%, Sigma Aldrich, St. Louis, MO, USA), poly(vinylpyrrolidone) (PVP, Sigma Aldrich, St. Louis, MO, USA) and nickel acetate tetrahydrate ((Ni(CH_3_COO)_2_•4H_2_O), 98%) were obtained from Sigma-Aldrich, St. Louis, MO, USA. Methylene blue (95.0%) was procured from LOBA Chemie, India, while ethanol and acetic acid were obtained from Scharlau, Spain.

### 2.2. Synthesis of NiTiO_3_/TiO_2_-Decorated CNFs

NiTiO_3_/TiO_2_-decorated CNFs were prepared via a sol-gel process, by adding 1.5 mL of titanium isopropoxide into 10 wt% solution of poly(vinylpyrrolidone) prepared in advance by dissolving 1.0 g of PVP in 10 mL deionized water at room temperature for 2 h duration under agitation. The poly(vinylpyrrolidone) solution was prepared as mentioned in previous reports [34,35,36,37,38,39]. The mixture was stirred until a transparent yellow gel was obtained. Then, nickel acetate tetrahydrate was added to this gel and continuously stirred until a green transparent gel was obtained. The prepared gel was fed into the plastic syringe of the lab-scale electrospinning device (NaBond Technologies Co., Limited, Hi Tech Park, Nanshan Dist., Shenzhen, Guangdong, China). The positive electrode was connected to the metallic end of the plastic syringe while the negative electrode was attached to the aluminum foil wrapped around the rotating cylinder. The voltage and distance between the positive and negative electrode during the spinning process were maintained at 20 kV and 15 cm, respectively. The NF mats formed on the surface of the aluminum foil were detached and placed in a vacuum dryer at a temperature of 50 °C for 24 h to get rid of the solvents. This was then calcined at 800 °C for 5 h. This synthesized material was named NiTiO_3_/TiO_2_-decorated CNFs. The TiO_2_@CNFs were prepared using a similar procedure except that the nickel acetate tetrahydrate was not added during gel preparation.

### 2.3. Catalyst Characterization

The morphological study of the NiTiO_3_/TiO_2_-decorated CNFs was performed by employing a scanning electron microscope (SEM Model JSM-5900, Japan Electron Optics Laboratory (JEOL Ltd., Tokyo, Japan) equipped with energy dispersive X-ray spectroscopic analysis (EDX). The crystallinity of the NFs was studied using an X-ray diffractometer (XRD, Rigaku Co., Tokyo, Japan) with Cu Kα radiation (λ = 1.54056 Å) in the range of 10° to 80° of two theta angles. A high-resolution transmission electron microscope (HRTEM) image was captured via JEOL Model JEM-2200FS, operated at 200 kV and equipped with EDX (JEOL Ltd., Tokyo, Japan).

### 2.4. Determination of MB Photodegradation

The MB photo-degradation in the presence of NiTiO_3_/TiO_2_-decorated CNFs was carried out in a simple batch photo-reactor consisting of a simple laboratory borosilicate glass bottle (150 mL). A visible fluorescent lamp was used as the light source (λ= 420–700 nm, I = 23–40 Wm^−2^ (Philips Co., Amsterdam, The Netherlands). The reactor (150 mL borosilicate glass bottle) was charged with 100 mL of MB aqueous solution with known concentration and a previously known amount of NiTiO_3_/TiO_2_-decorated CNFs photocatalyst. The solution was kept in the dark under continuous stirring for 20 min to determine the adsorption–desorption capacity of the photocatalyst towards MB. Later, the solution was stirred and exposed directly to the visible light radiation. The distance between the reactor and the lamp was 12 cm. The reaction temperature was controlled via a thermostat water bath. An aliquot of 3.0 mL of MB solution was withdrawn after specific time intervals under irradiation. The solution sample was centrifuged to separate the solid catalyst from the solution. Then, the filtered solution was introduced into a UV-visible spectrophotometer to determine the MB concentration present in the solution. The reaction parameters studied were MB concentration, reaction temperature, light intensity and the catalyst concentration. 

## 3. Results and Discussion

### 3.1. Catalyst Characterization

Figure 1A displays the SEM image of the electrospun nanofiber mat consisting of nickel acetate tetrahydrate, titanium isopropoxide, and poly(vinylpyrrolidone) after vacuum drying at 50 °C for 24 h. The image shows smooth and bead-free NFs. During the calcination process performed at 800 °C, NFs preserved their structure with the growth of tiny white NPs on the surface of NFs (Figure 1B). EDX analysis (Inset Figure 1B) indicates that the NFs mainly consisted of nickel (Ni), titanium (Ti), oxygen (O), and carbon (C), with the absence of any other element. 

Figure 2 displays the XRD patterns of the powder obtained after the calcination process. The results show the formation of TiO_2_ phases at 2θ of 27.02° which corresponds to (110) crystal plane; rutile phase (JCPDS # 00-004- 0551) at 2θ of 47.5°, 53.7°, 62.1°, 68.4° which agree with the (200), (105), (204), (220) plane, respectively; anatase phase (JCPDS # 21-1272). In addition, there is a formation of a hexagonal ilmenite NiTiO_3_ phase (JCPDS#01-039-12035) at 2θ of 24.4°, 35.5°, 40.6°, 43.7°, 50.9°, which match with (012), (110), (113), (202), (107), and (211) crystal planes, respectively. The low intensity peak observed at 2θ value of 32.4° is in agreement with the (002) plane of carbon-like graphite, which is formed due to the partial decomposition of carbon during the calcination process.

TEM EDX was performed to determine the chemical composition of synthesized NFs (Figure 3). Figure 3A displays the TEM image of a single selected NF along with the line TEM-EDX analysis and the corresponding EDX analysis is displayed in Figure 3B–E. It is evident from the figure that Ti, O, Ni, and C have the same distribution, which confirmed the formation of the NiTiO_3_-TiO_2_ composite. The carbon was covered with NiTiO_3_-TiO_2_ composite (Figure 3A). The presence of carbon may be enhancing the overall photocatalytic process through the enhanced photo-induced e^−^ and h^+^ separation, the more exposed area to the pollutant and the enhanced photodegradation rate.

### 3.2. Photocatalytic Degradation of MB

To understand the photodegradation process of MB dye using NiTiO_3_/TiO_2_-decorated CNFs catalyst, we considered the effect of MB dye concentration (C_i_), reaction temperature (T), photocatalyst dosage (NiTiO_3_/TiO_2_-decorated CNFs), and light intensity (I). The following mathematical model is designed to predict the photocatalytic reaction as a function of the studied parameters [40]:(1)$$K_{\mathrm{app}}\;=\;K^{\prime }\left(\frac{K_{R}}{1+K_{R}C_{i}}\right)\left(\exp\frac{-E_{a}}{\mathrm{RT}}\right)\left(\mathrm{mI}\right)\left(\frac{K_{\mathrm{NFs}}C_{\mathrm{NFs}}}{1+K_{\mathrm{NFs}}C_{\mathrm{NFs}}}\right)$$

Figure 4 shows the change of MB concentration versus irradiation time in the absence of photocatalytic NFs, and in the presence of TiO_2_@CNFs and NiTiO_3_/TiO_2_@CNFs. It can be observed from Figure 4 that the efficiency of MB photodegradation using NiTiO_3_/TiO_2_-decorated CNFs is higher than that of TiO_2_@CNFs. After 120 min of visible light irradiation, enhanced photodegradation of 82.4% and 62.5% is achieved for NiTiO_3_/TiO_2_-decorated CNFs and TiO_2_@CNFs, respectively. This achievement of 20% increase in photodegradation activity is significant. TiO_2_ alone is not very active under visible irradiation, however when TiO_2_ is doped with carbon, the band gap of titania narrows which makes it active under visible irradiation [3].

#### 3.2.1. Effect of Initial Dye Concentration (C_i_)

We studied the effect of initial MB dye concentration (5, 7.5, 10, and 15 mg L^−1^) on the photodegradation reaction under visible light irradiation in the presence of NiTiO_3_/TiO_2_-decorated CNFs. As seen in Figure 5A, the MB photodegradation decreases with increasing MB concentration, which may be plausibly due to the limited production of active radicals in the photocatalytic process at higher MB concentrations [30,41]. Furthermore, higher concentration of MB dye could absorb more light and cause the prevention of photons to reach to the surface of the photocatalyst, thereby reducing the efficiency of the photodegradation process [42,43,44]. Thus, the photodegradation process is efficient at low concentrations because of more available active sites on the surface of the photocatalyst to adsorb dye molecules. Photodegradation of organic compounds can be described through a pseudo-first order reaction according to the Langmuir Hinshelwood (LH) model (Equation (2)).
(2)$$r_{\mathrm{MB}}\;=\;-\frac{{\mathrm{dC}}_{f}}{\mathrm{dt}}\;=\;K_{1}C_{f}$$
(3)$$r_{\mathrm{MB}}\;=\;\frac{K_{\mathrm{LH}}K_{R}C_{f}}{1+K_{R}C_{i}}$$
(4)$$r_{\mathrm{MB}}\;=\;\frac{K_{\mathrm{LH}}K_{R}C_{f}}{1+K_{R}C_{i}}\;=\;K_{1}C_{f}$$
(5)$$K_{1}\;=\;\frac{K_{\mathrm{LH}}K_{R}}{1+K_{R}C_{i}}$$

Figure 5A shows the kinetics of the MB degradation over the NiTiO_3_/TiO_2_-decorated CNFs at various concentrations. The value of rate constant (K_1_) can be determined from the slope of the straight line obtained in Figure 5B. The value of K_1_ decreases from 0.0153 to 0.0044 min^−1^ with the increase of MB concentration from 5 to 15 mg L^−1^ (Table 1), which may be plausibly due to the limited formation of active radicals at higher concentrations. The relationship between K_1_ and C_i_ is obtained via a 1/K_1_ versus C_i_ plot (Figure 5B) and modified LH model by transforming Equation (6).
(6)$$\frac{1}{K_{1}}\;=\;\frac{1}{K_{\mathrm{LH}}}C_{i}\;+\;\frac{1}{K_{\mathrm{LH}}K_{R}}$$
to a straight-line equation y = ax + b, where



$x\;=\;C_{i}$





$a\;=\;1/K_{\mathrm{LH}}$





$K_{\mathrm{LH}}\;=\;1/a\;=\;1/17.595\;=\;5.683\times 10^{-2}$





$b\;=\;1/\left(K_{\mathrm{LH}}K_{R}\right)$





$K_{R}\;=\;1/(K_{\mathrm{LH}}\times b)\;=\;1/\left(5.683\times 10^{-2}\times 4.7345\right)\;=\;3.378$



#### 3.2.2. Effect of Reaction Temperature (T)

Figure 6A depicts the variation of photodegradation of MB versus irradiation time at various reaction temperatures. As shown in the figure, the photodegradation of MB increased with the increase in reaction temperature from 25 to 40 °C due to improved charge transfer at higher temperatures. The movement of electron–hole pairs becomes more active as the reaction temperature increases, which leads to the reaction between electrons, adsorbed oxygen and holes, and faster generation of hydroxyl radicals and hydroxyl ions, consequently improving the photodegradation of MB dye [45,46,47]. The temperature range of 20–80 °C is considered best for the effective photodegradation of organic compounds [45,48]. Further increases in the temperature could reduce the photocatalytic activity since the electron–hole recombination rate increased [45]. At the same time, low reaction temperature leads to reduction in the solubility of MB in water, causing partial condensation of MB in water [45,49]. 

Figure 6B shows the kinetic study of MB photodegradation over the NiTiO_3_/TiO_2_-decorated CNFs at various temperatures. The value of reaction rate (K_2_) increased from 0.0153 to 0.0222 min^−1^ with the increase of reaction temperature from 25 to 40 °C (Table 1), indicating an enhancement in the MB photodegradation efficiency with increasing temperature due to higher charge mobility. The rate constant (K_2_) can be determined from the slope of the straight line obtained in Figure 6B. The plot of K_2_ versus 1/T gives a straight-line relationship. We apply the Arrhenius equation Equation (7) to obtain the activation energy (E_a_) of the photocatalytic reaction.
(7)$$K_{2}\;=\;A\exp\frac{E_{a}}{\mathrm{RT}}$$

Applying the natural log (Ln) to both sides of Equation (7) removes the exponent to obtain Equation (8) as follows:(8)$${\mathrm{LnK}}_{2}\;=\;\left(\frac{E_{a}}{R}\right)\left(\frac{1}{T}\right)\;+\;\mathrm{LnA}$$

Transforming Equation (8) to a straight-line equation, y = ax + b, where



x = 1/T





$a\;=\;E_{a}/R$





$E_{a}\;=\;a\;\times \;R\;=\;2819.3\;\times \;8.314\;=\;2.344\;\times \;10^{4}$



#### 3.2.3. Effect of Light Intensity (I)

The photodegradation rate of organic compounds is affected by the light intensity and wavelength [42,50,51,52]. Figure 7A shows the influence of light intensity on the MB photodegradation. It is evident from the figure that the rate of MB photodegradation increased with the increase in light intensity. An increase in light intensity means the availability of a large number of photons, which improved the MB photodegradation. The rate constant (K_3_) can be determined from the slope of straight lines obtained from the plot of ln (C_i_/C_f_) versus time at various light intensities (Figure 7A). The value of K_3_ increased from 0.0153 to 0.0228 min^−1^ with the increase in light intensities from 25 to 40 Wm^−2^ (Table 1), which indicates an enhancement in the MB photodegradation with increasing light intensity. The correlation between K_3_ and I is obtained by plotting 1/K_1_ versus I (Figure 7B) and applying Equation (9) [40], since K_3_ is directly proportional to I as demonstrated in the following relationship: (9)$$K_{3}\;=\;\mathrm{mI}$$

Transforming Equation (9) to a straight-line equation, y = ax + b, where



x = I





a = m





$m\;=\;5\;\times \;10^{-4}$



#### 3.2.4. Effect of Photocatalyst Dose (NiTiO_3_/TiO_2_-Decorated CNFs)

Figure 8A shows the variation of MB concentration versus irradiation time at various NiTiO_3_/TiO_2_-decorated CNFs doses (200, 300, 400, and 500 mg L^−1^). As evident from the figure, the MB photodegradation increases with the increase in NiTiO_3_/TiO_2_-decorated CNFs dose, which may be plausibly due to increases in the available surface area of the NiTiO_3_/TiO_2_-decorated CNFs that improved the MB photodegradation. Furthermore, it is observed that the MB photodegradation at 400 mg L^−1^ is very close to the dose at 500 mg L^−1^. This could be plausibly due to decreases in the active sites resulting from the aggregation phenomenon at high doses of NiTiO_3_/TiO_2_-decorated CNFs. This also causes light scattering and screening effects that decrease the photoreactions and active radicals [30,53,54,55]. Figure 8B shows the kinetic study of MB photodegradation over NiTiO_3_/TiO_2_-decorated CNFs at various catalyst doses. The value of reaction rate (K_4_) increases from 0.0153 to 0.0324 min^−1^ with the increase in the photocatalyst dose from 200 to 500 mg L^−1^ (Table 1), indicating an improvement in the MB photodegradation because an increased NiTiO_3_/TiO_2_-decorated CNFs dose provides comparatively high active catalytic sites on the surface. The influence of NiTiO_3_/TiO_2_-decorated CNFs dosage on K_4_ is determined by employing a Langmuir-type relationship, Equation (10) [40,56]:(10)$$K_{4}\;=\;\frac{K_{o}K_{\mathrm{NFs}}C_{\mathrm{NFs}}}{1+K_{\mathrm{NFs}}C_{\mathrm{NFs}}}$$

This equation can be transformed to a straight-line equation, Equation (11), as follows:(11)$$\frac{1}{K_{4}}\;=\;\frac{1}{K_{o}K_{\mathrm{NFs}}}\left(\frac{1}{C_{\mathrm{NFs}}}\right)\;+\;\frac{1}{K_{o}}$$

The straight-line equation is y = ax + b, where



$x\;=\;1/C_{\mathrm{NFs}}$





$b\;=\;1/K_{o}$





$K_{o}\;=\;1/3.042\;=\;0.3287$





$a\;=\;1/K_{o}K_{\mathrm{NFs}}$





$K_{\mathrm{NFs}}\;=\;1/\left(17055\times 0.3287\right)\;=\;1.783\;\times \;10^{-4}$



Figure 9 shows the low- and high-magnification SEM images after using 500 mg L^−1^ catalyst dosage. As seen in the figure, the photocatalyst kept the nanofibrous structure intact after the photodegradation process. This shows the stability of the nanofibrous catalyst. 

#### 3.2.5. Development of Model Rate Equation 

The aforementioned results indicate that the rate constant is a function of MB concentration, reaction temperature, light intensity, and NiTiO_3_/TiO_2_-decorated CNFs dose according to Equations (6), (8), (9) and (11). The determined values of K_R_, E_a_, m, and K_cat_ employing multiple regression analysis are shown in Table 2. The equation constant k’ is calculated using Equation (2) by substituting the previously obtained values of C_i_, T, I, C_TiO2_ and values from Table 2. K_app_ can be rewritten as follows:(12)$$K_{\mathrm{app}}\;=\;K^{\prime }\left(\frac{3.378}{1+3.378\times 5}\right)\left(\exp\frac{-2.344\times 10^{4}}{8.314\times 298}\right)\left(5\times 10^{-4}\;\times 25\right)\left(\frac{1.78\times 10^{-4}\times 200}{1-1.78\times 10^{-4}\times 200}\right)$$

From Figure 2, K_app_ = reaction rate constant = 0.0153.
$$\begin{array}{l} 0.0126\;=\;K^{\prime }\left(\frac{3.378}{1+3.378C_{i}}\right)\left(\exp\frac{-2.344\times 10^{4}}{8.314T}\right)\left(5\times 10^{-4}I\right)\left(\frac{-1.78\times 10^{-4}C_{\mathrm{CNF}}}{1-1.78\times 10^{-4}C_{\mathrm{CNF}}}\right)\\ K^{\prime }\;=\;-1.857908\times 10^{6}\\ K_{\mathrm{app}}\;=\;-1.857908\times 10^{6}\left(\frac{3.378}{1+3.378C_{i}}\right)\left(\exp\frac{-2.344\times 10^{4}}{8.314T}\right)\left(5\times 10^{-4}I\right)\left(\frac{-1.78\times 10^{-4}C_{\mathrm{CNF}}}{1-1.78\times 10^{-4}C_{\mathrm{CNF}}}\right) \end{array}$$

A comparison between the experimental data and the determined K_app_ for the MB photodegradation at various conditions is shown in Figure 10. The obtained plot shows that the experimental data are in good agreement with the model-calculated data, which confirms the authenticity of our model to predict the reaction rate constant at various operational conditions. 

### 3.3. Photocatalytic Mechanism

The photocatalytic mechanism is proposed for the NiTiO_3_/TiO_2_-decorated CNFs in the photodegradation of MB dye. The band alignment and charge transfer diagram of the NiTiO_3_/TiO_2_-decorated CNFs is shown in Figure 11. The photocatalytic process is responsible for the production of active radicals and ions. However, the higher bandgap of TiO_2_ is responsible for the production of fewer active radicals in visible light [57]. 

The separation of the photogenerated electron–hole pairs across the NiTiO_3_/TiO_2_-decorated CNFs may be ascertained by calculating the conduction band (CB) and valence band (VB) potentials of the components. Specifically, these energies are calculated using the following empirical formulae:(13)$$E_{\mathrm{CB}}\;=\;\chi -E^{e}-0.5E_{g}$$
(14)$$E_{\mathrm{VB}}\;=\;E_{\mathrm{CB}}\;+\;E_{g}$$
where the VB potential is denoted by E_VB_, and the CB potential is denoted by E_CB_. E^e^ is the energy of free electrons versus NHE, which is 4.5 eV [57,58]. E_g_ is the band gap energy of the semiconductor. Finally, the electronegativity of the semiconductor is denoted by the letter χ.

The conduction band (CB) and valence band (VB) potentials of NiTiO_3_ with respect to the standard hydrogen electrode (SHE) are 0.23 eV and +2.62 eV, respectively [58]. The NiTiO_3_ has smaller bandgap (2.62 eV) than pure TiO_2_ (3.2 eV), thus the VB position of NiTiO_3_ is higher than TiO_2_ (+2.7 eV), which enables the hole transfer from the VB of NiTiO_3_ to the VB of TiO_2_. The resulting photogenerated hole (h^+^) reacts with H_2_O/OH^-^ to produce hydroxyl radicals [30,59], while CB of NiTiO_3_ is located at a lower position than that of TiO_2_. As light irradiates, the electrons in the VB of NiTiO_3_ get excited and reach CB, which results in partial vacancy in the VB, and the electrons from the CB of TiO_2_ get transferred to the CB of NiTiO_3_. The role of NiTiO_3_ in the photocatalytic mechanism suggests that it can be used as an efficient sensitizer under visible light. The photogenerated electrons in NiTiO_3_ move freely towards the surface of the CNFs, suggesting the low recombination of photogenerated electrons and holes [60]. The oxygen molecules react with the photogenerated electrons in the CB of NiTiO_3_ to produce ^•^O^2−^ without recombining with the holes present on the surface of TiO_2_ [36]. The resulting ^•^O^2−^ react with the hydrogen ion and produce HOO^•^, which consequently alter the MB molecule in the solution. The achieved photocatalytic activity of the NiTiO_3_/TiO_2_ heterojunction is higher than that of the pure TiO_2_ [61]. The photodegradation mechanism of MB using NiTiO_3_/TiO_2_-decorated CNFs can be explained as follows [1,30]:


*In the conduction band:*


e^−^ + O_2_
→ O_2_^−•^

O_2_^−•^ + H_2_O → HO_2_^•^ + OH^−^

HO_2_^•^ + H_2_O → H_2_O_2_ + HO^•^

H_2_O_2_
→ 2OH^•^


*In the valence band:*


h^+^ + OH^−^ → OH^•^

OH^•^ + MB dye → Degradation products (CO_2_+H_2_O).

### 3.4. Catalyst Recyclability Data

The stability and recyclability of the synthesized NiTiO3/TiO_2_-decorated CNFs catalyst is tested for MB photodegradation. Figure 12 shows the catalyst recyclability data of the catalyst after 120 min of visible light irradiation for four consecutive runs. The photodegradation activity of the catalyst showed very little decrease (~1.3%) after four runs. The data show that the catalyst is stable and recyclable. 

### 3.5. Comparison of Our Results with the Literature

The results obtained from the photocatalytic degradation of MB in the presence of catalyst NiTiO_3_/TiO_2_-decorated CNFs is compared with the catalysts reported in the literature with titanate as an active ingredient. The results are reported in Table 3. Marco Alejandro Ruiz Preciado demonstrated the photocatalytic activity of NiTiO_3_ for the photodegradation of MB using thin films under visible light [62]. Their study determined a reaction rate constant of 0.0030 min^−1^. Dao et al. synthesized co-doped NiTiO_3_/g-C_3_N_4_ composite photocatalysts and used them for MB photodegradation under visible light irradiation [63]. The combination of co-doped NiTiO_3_ and g-C_3_N_4_ enhanced the photocatalytic performance of the composite photocatalyst. The co-doped composite photocatalyst (1% co-doped NiTiO_3_/g-C_3_N_4_) produced a higher k_app_ value (0.0072 min^−1^), while the one loaded with 3% co-doped NiTiO_3_/g-C_3_N_4_ showed a lower value of the rate constant (0.0057 min^−1^). Kitchamsetti et al. synthesized MnTiO_3_ perovskite nanodiscs and utilized them for the photocatalytic degradation of several organic dyes [64]. These nanodiscs provide stable and recyclable photocatalytic activity under Xenon lamp irradiation. The results showed 89.7, 80.4, 79.4, and 79.4% degradation of MB, rhodamine B, Congo red, and methyl orange at rate constants of 0.011, 0.006, 0.007, and 0.009 min^−1^, respectively. Sadjadi et al. reported the synthesis of NiTiO_3_ NPs loaded on MCM-41 and used them for the photocatalytic degradation of MB under UV and visible light irradiation [65]. The results showed that the NiTiO_3_ NPs/MCM-41 composite has higher photocatalytic activity than that of NiTiO_3_ NPs and exhibited a kinetic rate constant of 0.018 min^−1^. Khan et al. synthesized one-dimensional NiTiO_3_ NFs via electrospinning method and loaded them with acetic acid-treated, exfoliated and sintered sheets of graphitic carbon nitride (AAs-gC_3_N_4_), which produced a unique heterogeneous structure [66]. The weight ratio of NiTiO_3_ NFs to porous AAs-gC_3_N_4_ was 40:60, and displayed very good photodegradation of MB at a rate constant of 0.0310 min^−1^. This comparison clearly showed that the NiTiO_3_/TiO_2_-decorated CNFs catalyst possesses a high kinetic rate constant (0.0324 min^−1^) compared to other reported catalysts.

## 4. Conclusions

Electrospinning and calcination methods are utilized in the successful synthesis of NiTiO_3_/TiO_2_-decorated CNFs. The chemical composition, structural information, and phase analysis are evaluated via characterization techniques which confirm the fabrication of CNFs. Several operational parameters for the photodegradation application, such as the effect of the initial dye concentration, reaction temperature, light intensity, and catalyst dose, are considered. A successful model is achieved through the determination of reaction rate constant (k_app_) to predict the photocatalytic activity of the composite NiTiO_3_/TiO_2_-decorated CNFs towards photodegradation of MB dye under visible light in a batch reactor. The photodegradation is 82.4% and 62.5% for NiTiO_3_/TiO_2_-decorated CNFs and TiO_2_@CNFs, respectively, after 120 min visible light irradiation. The reaction constants K_1_, K_2_, K_3_, and K_4_ determined for initial dye concentration, reaction temperature, light intensity, and photocatalyst dose, respectively, are in full agreement with the estimated results from the model for the rate constant (k_app_). The results suggest further exploration of the electrospinning fabrication method for the synthesis and use of NiTiO_3_/TiO_2_-decorated CNFs towards effective photocatalytic activity and future applications in environmental remediation. The recyclability data of the synthesized NiTiO_3_/TiO_2_-decorated CNFs catalyst show that the catalyst is quite stable, and a very small decrease is observed in the MB photodegradation activity from 82.4 to 81.1% after 120 min of visible light irradiation for four consecutive runs. The data also showed good recyclability characteristics of the catalyst. The performance of our catalyst for MB photocatalytic degradation is compared with those reported in the literature and the NiTiO_3_/TiO_2_-decorated CNFs exhibit higher rate constant compared to other catalysts.

## Figures and Tables

**Figure 1 polymers-15-00109-f001:**
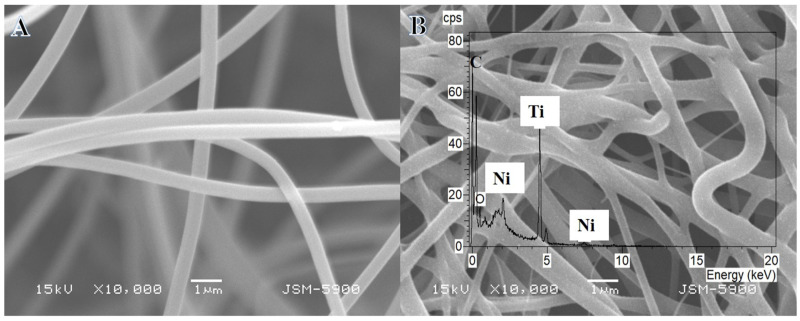
SEM images of the electrospun NiAc/TiIP/PVP nanofiber mats after drying at 50 °C overnight (**A**) and the produced NiTiO_3_/TiO_2_-decorated CNFs after calcination in argon at 800 °C (**B**).

**Figure 2 polymers-15-00109-f002:**
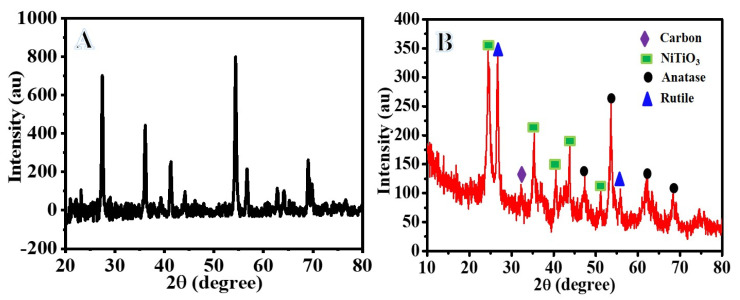
XRD patterns of the produced powder after calcination of electrospun TiIP/PVP (**A**) and NiAc/TiIP/PVP (**B**) nanofiber mats at 800 °C in argon atmosphere.

**Figure 3 polymers-15-00109-f003:**
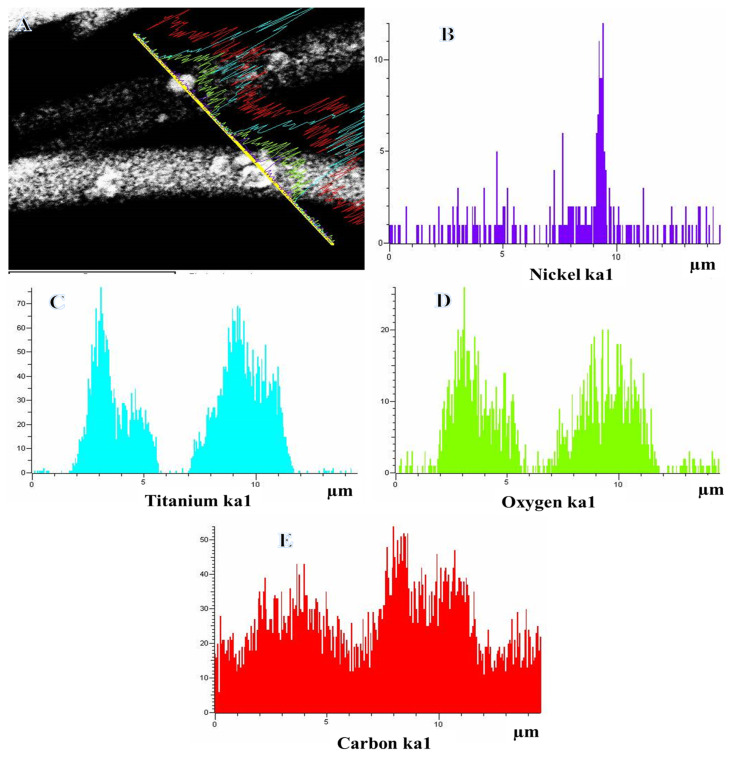
TEM image for a single calcined nanofiber along with the line TEM EDX analysis (**A**) and the corresponding Ti, C, O and Ni line analyses TEM EDX (**B**–**E**).

**Figure 4 polymers-15-00109-f004:**
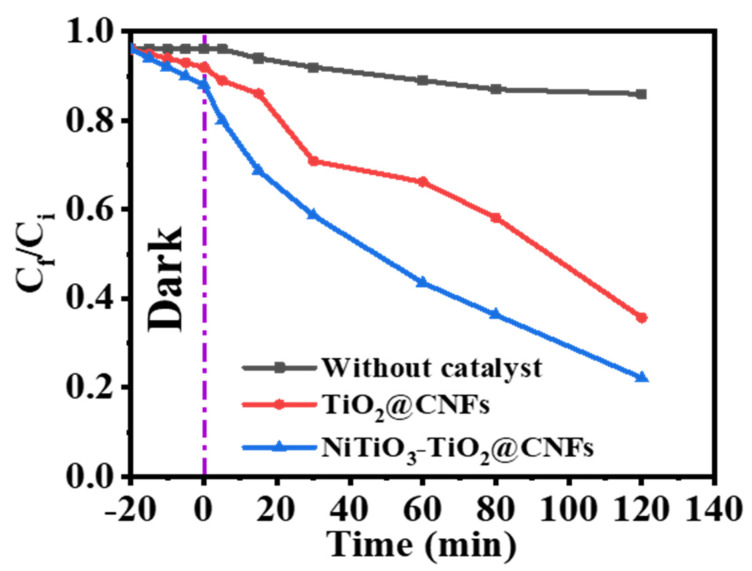
The photodegradation profile of MB dye. The parameters are: photocatalyst amount = 0.2 gm L^−1^, C_i_ = 0.1 M, T = 298 K, and I = 25 W m^−2^.

**Figure 5 polymers-15-00109-f005:**
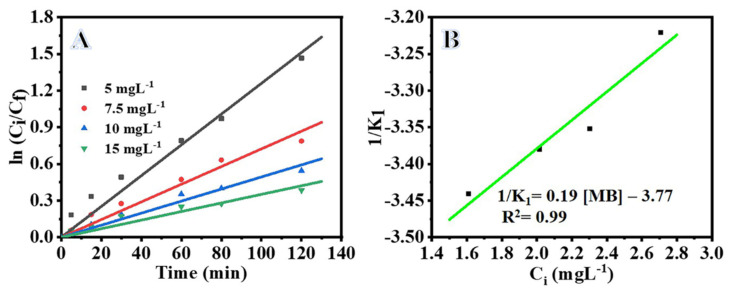
Influence of [MB] on photodegradation of MB, (**A**) ln Ci/Cf vs. time (**B**) modified LH plot for MB photodegradation. The parameters are: photocatalyst amount = 0.2 gm L^−1^, C_i_ = 0.1 M, T = 298 K, and I = 25 W m^−2^.

**Figure 6 polymers-15-00109-f006:**
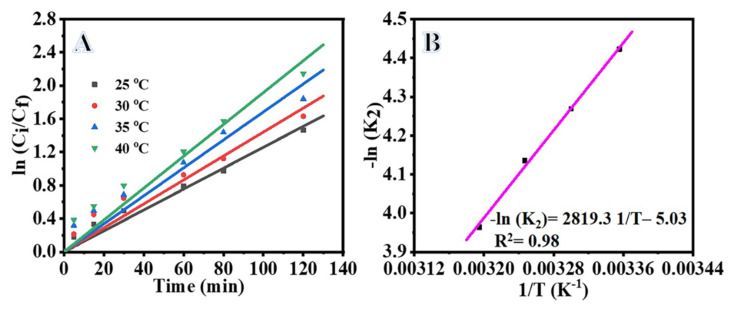
Influence of reaction temperature on photodegradation of MB (**A**), ln Ci/Cf vs. time (**B**), Arrhenius plot for MB photodegradation. The parameters are: photocatalyst amount = 0.2 gm L^−1^, C_i_ = 0.1 M, and I = 25 W m^−2^.

**Figure 7 polymers-15-00109-f007:**
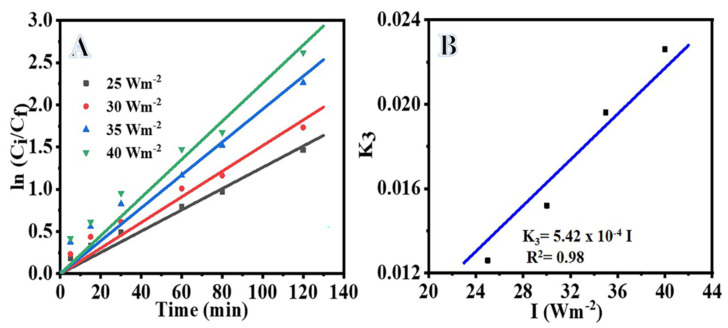
Influence of light intensity on photodegradation of MB, (**A**) ln C_i_/C_f_ vs. time (**B**) plot of K_3_ vs. I. The parameters are: photocatalyst amount = 0.2 gm L^−1^, C_i_ = 0.1 M and T = 298 K.

**Figure 8 polymers-15-00109-f008:**
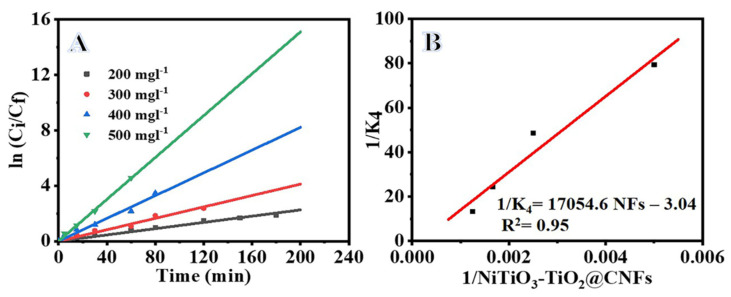
Influence of catalyst amount on photodegradation of MB (**A**), ln C_i_/C_f_ vs. time (**B**), Langmuir-type plot for photodegradation of MB. Parameters are: C_i_ = 0.1 M, T = 298 K, and I = 25 W m^−2^.

**Figure 9 polymers-15-00109-f009:**
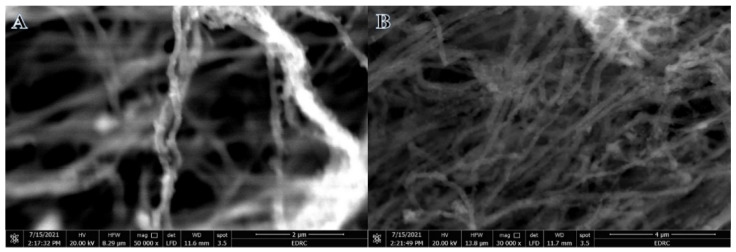
SEM images of the used photocatalyst. (**A**) 50,000 magnification, (**B**) 30,000 magnification.

**Figure 10 polymers-15-00109-f010:**
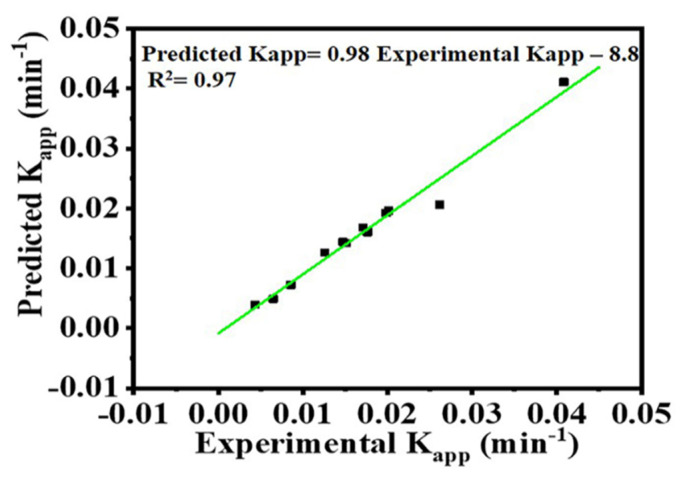
Comparison of experimental K_app_ and predicted K_app_ values.

**Figure 11 polymers-15-00109-f011:**
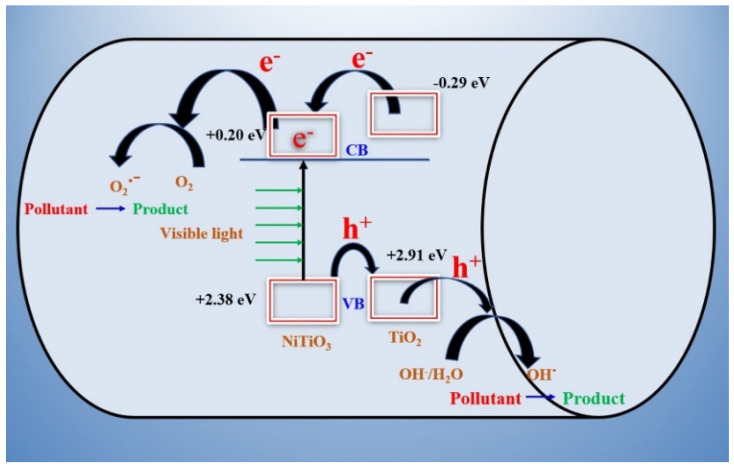
Schematic diagram for the creation and influence of electrons and holes in the photocatalytic degradation of MB.

**Figure 12 polymers-15-00109-f012:**
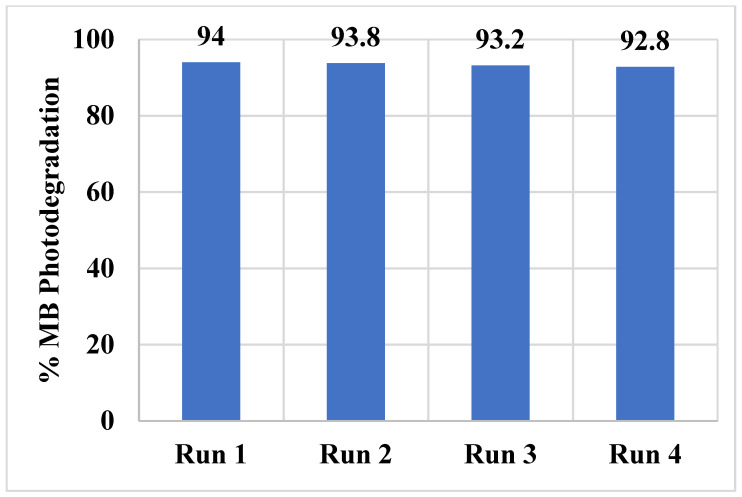
Catalyst recyclability data of NiTiO3/TiO_2_-decorated CNFs after 120 min of visible light irradiation.

**Table 1 polymers-15-00109-t001:** Rate constants of reactions of MB photodegradation at various MB concentrations, reaction temperatures, light intensities, and NiTiO_3_/TiO_2_-decorated CNFs catalyst doses.

MB Dye Concentration (mg L^−1^)	Rate Constant (min^−1^) K_1_
5	0.0153
7.5	0.0088
10	0.0068
15	0.0044
Reaction temperature (°C)	Rate constant (min^−1^) K_2_
25	0.0153
30	0.0166
35	0.0188
40	0.0222
Light intensity (W/m^2^)	Rate constant (min^−1^) K_3_
25	0.0153
30	0.0166
35	0.019
40	0.0228
Catalyst dose (mg L^−1^)	Rate constant (min^−1^) K_4_
200	0.0153
400	0.0216
600	0.027
800	0.0324

**Table 2 polymers-15-00109-t002:** Constant values obtained using multiple regression analysis for model equation.

Parameter	k′	K_R_ (L mg^−1^)	E_a_ (J mol^−1^)	R (J K^−1^ mol^−1^)	m (m^2^ W^−1^ min^−1^)	K_NFs_ (L mg^−1^)
Values	3.7551 × 10^4^	4.99	1.9204 × 10^4^	8.314	6 × 10^−4^	2.444 × 10^−3^

**Table 3 polymers-15-00109-t003:** Performance comparison of our catalyst with those reported in the literature.

Catalyst Used	Rate Constant (min^−1^)	References
NiTiO_3_ NFs	0.0030	[62]
1%Co-NiTiO_3_/g-C_3_N_4_	0.0072	[63]
3%Co-NiTiO_3_/g-C_3_N_4_	0.0057	[63]
MnTiO_3_ perovskite nanodiscs	0.0110	[64]
NiTiO_3_ NFs/MCM-41	0.0189	[65]
NiTiO_3_ NFs/40wt% AAs-gC_3_N_4_	0.0310	[66]
NiTiO_3_/TiO_2_-decorated CNFs	0.0324	Our work

## Data Availability

The data presented in this study are available from the corresponding author upon reasonable request.

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
