# Peer review of "Enhanced Organic Pollutant Removal Efficiency of Electrospun NiTiO3/TiO2-Decorated Carbon Nanofibers"

_polymers, 2022, doi:10.3390/polym15010109_

Round 1
Reviewer 1 Report
In this manuscript, the authors reported organic pollutant removal of electrospun NiTiO3/TiO2-decorated carbon nanofibers. The authors combined NiTiO3/TiO2 with electrospun CNFs to synthesize the composites, and tested their photocatalytic performances. However, I do not recommend the publication of this work in Polymers at this stage for the following reasons:
1. A major concern is that the novelty of this work should be addressed, especially the performance superiority when compared with other types of materials. The development of NiTiO3/TiO2-coated CNFs should be included by authors with typical comparative results.
2. The introduction paragraph should be refined with a clear structure. The excellent properties of various functional nanoparticle-based photocatalysis (e.g., ZnO, BiClO, BiBrO) are suggested to be contained, especially the recent advances.
3. In Figure 3, the element line scanning and the texts are hard to read, which need to be improved. The authors can use elemental mapping of one single CNF for better explanation.
4. Further material characterizations are needed, including FT-IR, UV-Vis diffuse reflectance of the samples. SEM and XRD analysis of the samples after cycling tests should also be added to compare with the initial states.
5. Except for the photodegradation, the profile in dark should be added to show its adsorption property. This is also a prerequisite before conducting photodegradation.
6. It is necessary to measure K-values after adding various scavengers in MB degradation. In general, EPR tests of the samples are required to better demonstrate the mechanism and property. Based on these results, Figure 10 needs to be improved for better demonstration.
Author Response
Dear Reviewer,
Thank you for your kind response about our manuscript (Polymers-2048406) entitled “Enhanced Organic Pollutant Removal Efficiency of Electrospun NiTiO3/TiO2-Decorated Carbon Nanofibers”. The comments were helpful to strengthen and restructuring the manuscript. We have now modified the manuscript according to the given comments. To facilitate following the changes in the revised manuscript, we have marked them in yellow color. We hope our responses will address all the comments in a comprehensive manner. It will be our pleasure to respond further in case more clarifications are required.
Thank you very much.

Reviewer 2 Report
Review report
Manuscript title: Enhanced Organic Pollutant Removal Efficiency of Electrospun NiTiO3/TiO2-Decorated Carbon Nanofibers
Manuscript ID: polymers-2048406
Review comments: The work is interesting. The work demonstrates about the Enhanced Organic Pollutant Removal Efficiency of Electrospun NiTiO3/TiO2-Decorated Carbon Nanofibers. I think it will be published in the current journal after modification the issues raised.
1. In the abstract there are mixing of present and past form of tenses. Its totally wrong. It should be one particular form.
2. There is a wrong choosing of keywords. One word should be used as one word or maximum two.
3. In figure 10, the schematic image needs to be more clear.
4. The ref list needs to be MDPI styles.
5. There are many grammatical mistakes. The English needs to be rechecked and make it more corrections.
Author Response

(The authors gave the same response as above.)

Reviewer 3 Report
Current manuscript entitled “Enhanced Organic Pollutant Removal Efficiency of Electrospun NiTiO3/TiO2-Decorated Carbon Nanofibers” by “Maafa et al” deliberated on the preparation of heterojunction NiTiO3/TiO2-decorated CNFs and their visible-light-driven MB degradation as a model dye. The photocatalyst was prepared using the electrospinning technique followed by calcination at 800ºC of the electrospun nanofiber mats compose of titanium (IV) isopropoxide, nickel acetate tetrahydrate and poly(vinylpyrrolidone). The results obtained favor effective photodegradation efficiency and are in good agreement with the modeling results. The work seems good and can be accepted after addressing the following comments.
1. The authors should revise the introduction, logical way of presentation is missing.
2. Provide the information of the electrospinning device.
3. Provide the XRD for all the synthesized materials
4. Put the SEM images at different magnifications
5. Figure 3 is not clear.
6. Incorporate some more information on the Influence of reaction temperature on photodegradation of MB.
Author Response

(The authors gave the same response as above.)

Round 2
Reviewer 1 Report
I am satisfied with your response. Good luck.Reviewer 3 Report
The revised manuscript can be accepted for publication